# Subjective Perceptions and Their Characteristics of Middle School Students Regarding the Effectiveness of the "0th Period Physical Education Class" in South Korea: The Q Methodology Application

**Wonjae Jeon [1], Goomyeung Kwon [1,\*] and Kihong Joung [2,\*]**

[1] Department of Senior Sports Course, Daegu Haany University, Gyeongsan-si 38610, Korea; dreamj007@hanmail.net

[2] Department of Continuing Education Center and Physical Education, Kangnam University, Yongin-si 16979, Korea

\* Correspondence: kgm@dhu.ac.kr (G.K.); King@kangnam.ac.kr (K.J.); Tel.: +82-10-3691-9079 (G.K.); +82-10-5414-6978 (K.J.)

**Abstract:** The aim of this study is to explore the subjective perception types and characteristics of Korean middle school students regarding participation in the "0th period physical education class", a class involving physical movement that takes place before the start of regular school classes in the morning. This goal was achieved by applying the Q methodology, which can categorize the subjective viewpoints of research participants. The selection of the final 25 Q-samples was done by composing the Q-population. Twenty middle school students were selected as the P-sample, and Q-sorting was performed on them. The PQ method program (version 2.35) was used to perform centroid factor analysis and varimax rotation. The study presented five types with a total variance of 87%. Types 1 to 5 (N = 4, 4, 4, 5, and 3) pertained to a potent means of enhancing lesson concentration and academic performance, efficient activities to improve physical ability and a healthy body image in adolescence, the motivating power behind a stable school life and sociability development, building an upright character and successful changes in daily life, and raising awareness of the importance of participating in sports and the importance of physical activity, with eigenvalues (EVs) of 3.89, 4.48, 3.96, 5.16, and 2.58, respectively, and explanatory variances of 0.10, 0.22, 0.13, 0.33, and 0.09, respectively. Moreover, consensus statements for each factor were demonstrated as being Q24 and Q25. The findings in this study supported the academic foundation for the official introduction and activation of "0th period physical education classes" in the Korean education community for the near future.

**Keywords:** 0th period physical education class; Q methodology; subjectivity; Korean physical education policy; Korean education policy; P.E.

## 1. Introduction

More than 20 years ago, school violence emerged as a serious social problem in Korea. In particular, in a more recent case occurred on 20 December 2011, when a middle school student, aged 13, committed suicide in his apartment in Daegu, South Korea. The decisive reason for choosing suicide was due to bullying from two other students. This incident sent the whole nation into a state of shock, and then the Korean government began to develop countermeasures [1].

It is worth noting that physical education (P.E.) is at the center of various policies proposed by the government for solving the problem of school violence. Representatively, there was a school sports club activity implemented in October 2011 as part of the "Comprehensive Measures to Eradicate School Violence Policy" [2]. In addition, the educational

value of school sports club activities that play a large role in preventing school violence is being proven [3].

In Korean society, P.E. was neglected in school education [4]. Even so, as of late, the importance of P.E. in Korean schools has been rekindled. This is due to the value of school sports being highlighted and becoming visible as a way of preventing and healing various social problems that arise from the Korean school field, such as the excessive competition in entrance exams, school violence, and suicide [5]. As such, P.E. has become an essential key for the successful educational value of students in Korea.

There is another reason why school P.E. was marginalized from school education in the past. As mentioned earlier, it is because of the atmosphere created among the parents of Korean society that academic performance is more important than the physical activity of children. Since Korean society has boundless passion regarding education, many parents prioritize studying over their children's physical activities [6]. This perspective of the Korean education system is one of the main factors that hinder the expansion of physical activities for youth.

In the educational system of Korean society, the academic implications of early morning exercise began being shared among the Korean sports community as the prominent effect of the "0th period P.E. class", which was broadcast via mass media. Moreover, the Korean education community has begun to show interest in the effectiveness of the class. It started in 2005 at Naperville Central High school in the state of Illinois in the USA. The early morning exercise has proven, scientifically, that doing physical activities before beginning regular class in the early morning contributes to brain balance and function, which then improve concentration, memory, and attitude in regular classes [7]. As such, academic interest in schools has been more noticeable in Korea since the media first reported on the case of the school that conducted the "0th period P.E. class" through MBC's 2010 "Brain Revolution Project" [8].

The title of "0th period P.E. class" has been used in various ways in domestic and foreign studies. The class is defined as team sports such as soccer and basketball or personal sports such as weight training, in which students participate before their morning classes begin (around 7 a.m. or 7:30 a.m. to 8:30 a.m.) [8].

Looking at the term in detail, it is used with various names such as "zero hour class physical activity", "0th period physical activity", "early morning physical activities", and "early morning physical activity" [4,5,9–11]. Based on these media reports and research results, there has recently been a movement in Korean society to introduce this "0th period P.E. class" as an official curriculum. Nevertheless, this class is not currently included in the official system of the school education system [12]. In addition, some schools are autonomously implemented according to the will of the principal. In other words, despite the multifaceted effectiveness of this class, it has not actively been introduced in the Korean school field. As has been noted by a variety of scholars, "0th period P.E. class" must be an example that can provide positive implications from various perspectives for the future of the public education system in Korea [5,11,13].

More recently, numerous studies have tried to find and explore the effect of the "0th period P.E. class". Much scholarly work has been conducted on the topics of perception, participation experience, physical strength, educational value, and school life adaptation and satisfaction [12]. Kim and Kang [14] argued that 0th-period physical activity has a positive effect on middle school students' learning attitude, physical strength, and physical self-efficacy. Jung et al. [15] proved that Korean middle school students' participation in P.E. activities in the 0th period plays a helpful role in improving learning through increased concentration. When looking at overseas research regarding early morning physical activities, Schumacher et al. [11] stated that, among adults with obesity, there are benefits of consistent morning exercise for both increasing exercise and enhancing weight management. As such, despite the progress of research on the "0th period P.E. class", there is no mention of the actual perception structure of participating students made in account of the explanation of previous studies. Namely, research on the perceptions experienced

and felt by students participating in the early morning physical activities has received relatively little attention from Korean scholarship.

Contrasting the deductive approach for verifying the theoretical hypotheses adopted by prior studies, the present study aims to examine the effectiveness of the "0th period P.E. class" acknowledged by and from the subjective perspective of Korean middle school students. Regarding this, the Q methodology is applied in this study because it is known as a useful method that can objectively derive the subjectivity of research participants. Therefore, this study's purpose is to analyze in-depth the experiences of students participating in the "0th period P.E. class" and further discover various educational meanings with regard to the trend and successful settlement of the class. These findings could provide an academic perspective on introducing and implementing policy directions in which the "0th period P.E. class" can be established within the Korean public education system.

The present study proposes the following research questions: How is the subjective perception structure of Korean middle school students regarding participation in the "0th period P.E. class" categorized? What are the characteristics and implications of each categorized type?

## 2. Methods

The Q methodology was developed by William Stephenson in 1953 and is a method that can categorize the subjective viewpoints of research participants [16]. In contrast to the R methodology, the Q methodology is not an operational definition of the researcher, but rather an operant methodology that represents perception in the participant's own thoughts and language [17]. Moreover, the Q methodology is a way of exploring human-aware subjectivity and is appropriate for research for revealing personal experiences, preferences, values, and beliefs about various research topics [18]. Thus, it is an appropriate method for investigating the perception of middle school students with regard to the "0th period P.E. class" [19].

### 2.1. Q-Population (the Concourse of Statements) and Q-Sample

The first step in the Q methodology is constructing a Q-population. The Q-population is a collection of all possible statements (the "concourse") on the subject of the study (these statements are normally collected by means of literature analysis, group interviews, in-depth interviews, etc.) [20]. This allows for the adoption of statements focusing on subjective perspectives, emotions, attitudes, etc. [21]. Therefore, the Q-population was constructed using a total three-dimensional method [22]. First, an FGI (focus group interview) was conducted with 3 students who participated in 0th period P.E. classes during their 3 years of middle school, 2 professors who majored in sports pedagogy, and 2 professors who majored in the sociology of sports. Second, in-depth interviews with each student were conducted. The FGI and in-depth interviews had a duration of about 60–80 min and were performed twice. Third, we carried out a literature review and document analysis related to the 0th period P.E. class. We finally compiled 41 statements (the "concourse") through the above process.

The 41 statements were sorted and then thematically grouped. In particular, repetitive statements were removed, while unclear statements were reworded for clarity [23]. Peer reviews were then conducted of the statements by colleagues unrelated to the project to check for clarity, conciseness, and to identify any potential themes that may have been omitted [24]. The final Q statements are presented in Table 1. In addition, a reliability test was conducted by performing Q-sorting twice for 3 respondents. The result was derived using SPSS Statistics 22.0 with $r = 0.71$, which indicated sufficient reliability [25].

**Table 1.** Q statements.

| Q Number | Q Statements |
|---|---|
| 1 | It becomes a medium with which to prevent bullying and school violence. |
| 2 | It can boost patience and promote the spirit of competition. |
| 3 | It can actively alter a child's personality. |
| 4 | It can alleviate stress and lower depression. |
| 5 | It can encourage close relationships/connections with instructors or teachers. |
| 6 | It could be a means of reviving and enhancing friendships. |
| 7 | It can foster a sense of belonging within the school. |
| 8 | It can make life at school more pleasant and captivating. |
| 9 | It is one of the key ways in which to learn about consideration and regard for others. |
| 10 | It can refine muscular power and stamina. |
| 11 | It can refine flexibility and cardiovascular stamina. |
| 12 | It can improve motor skills such as dexterity and swiftness. |
| 13 | It is useful in controlling body weight, averting obesity, and dieting (weight loss). |
| 14 | It encourages interest in the participation of various sports events. |
| 15 | It can aid children in recognizing the positive attributes of physical activity. |
| 16 | It can promote more active participation in other classes aside from physical education classes. |
| 17 | It can make students feel less bored in other classes aside from physical education classes. |
| 18 | It can aid in improving the memory of the learning content. |
| 19 | It can induce an improvement in class concentration. |
| 20 | It can induce an improvement in academic performance. |
| 21 | It can inspire a sense of responsibility and achievement for students. |
| 22 | It leads to the acquisition of an intended life habit. |
| 23 | It can build confidence in various fields and can improve self-esteem. |
| 24 | It leads to healthier eating habits. |
| 25 | It increases curiosity in sports, making it possible to continue into adulthood. |

### 2.2. P-Sample

Brown [19] suggested to select the P-sample (participants) by centering on those who have an interest, known views, and are experts in relation to the research topic. The Q methodology is a method for measuring subjective opinions within an individual, and there is no limit to the number of P-samples because inter individual difference in significance is the core of the methodology rather than inter individual difference [24]. Furthermore, the number of P-samples should be fewer than the number of items in the Q-sample for statistical reasons [17]. This is because the population's characteristics are not inferred from the characteristics of the P-sample [26]. With this in mind, the P-sample was selected and made up of 20 Korean middle school students who have participated in the "0th period P.E. class" for approximately 2 years. Detailed P-sample characteristics and factor weights are shown in Table 2.

**Table 2.** Summary of characteristics for the P-sample and factor weights.

| Factor (Type) | P-Sample | Grade | Gender | Education Period (Years) | Activities | Factor Weight |
|---|---|---|---|---|---|---|
| Type 1. (N = 4) | 6 | 2nd | Male | 1.6 | Soccer & Basketball | 0.68 |
| | 7 | 3rd | Male | 2.6 | Weight training | 0.59 |
| | 13 | 3rd | Female | 2.6 | Volleyball | 0.90 |
| | 14 | 3rd | Female | 2.6 | Circuit weight training | 0.88 |
| Type 2. (N = 4) | 1 | 3rd | Female | 2.6 | Soccer | 0.75 |
| | 8 | 3rd | Male | 2.6 | Baseball | 0.80 |
| | 15 | 2nd | Female | 2 | Soccer & Basketball | 0.84 |
| | 19 | 2nd | Male | 2 | Circuit weight training | 0.68 |
| Type 3. (N = 4) | 2 | 3rd | Male | 2.6 | Basketball | 0.90 |
| | 9 | 3rd | Male | 2.6 | Soccer | 0.92 |
| | 16 | 3rd | Male | 2.6 | Baseball | 0.80 |
| | 20 | 3rd | Male | 2.6 | Soccer | 0.71 |

<div align="center"><b>Table 2.</b> <i>Cont.</i></div>

| Factor (Type) | P-Sample | Grade | Gender | Education Period (Years) | Activities | Factor Weight |
|---|---|---|---|---|---|---|
| Type 4. (N = 5) | 3 | 3rd | Female | 2 | Volleyball | 0.60 |
| | 4 | 3rd | Male | 2 | Circuit weight training | 0.59 |
| | 10 | 3rd | Female | 2.6 | Basketball | 0.54 |
| | 17 | 2nd | Male | 2 | Volleyball | 0.64 |
| | 18 | 2nd | Male | 2 | Soccer | 0.89 |
| Type 5. (N = 3) | 5 | 3rd | Female | 2.6 | Basketball | 0.87 |
| | 11 | 3rd | Female | 2.6 | Circuit weight training | 0.59 |
| | 12 | 3rd | Female | 2.6 | Volleyball | 0.79 |

### 2.3. Q-Sorting and Factor Analysis

Brown [27] noted that Q-sorting means that the P-sample favorably evaluates a previously selected Q-sample. More specifically, after the participants carefully read and understood the Q-samples, they arranged the Q-sorting table from the most agreed with (+4) to the most disagreed with (−4) questions (Figure 1) according to their thoughts. The Q-sorting table was constructed on a 9-point scale (−4 to +4).

The Q-sorting method is split into a forced sorting method and an unforced sorting method. Between them, the former was employed. This method requires the researcher to set the number of responses in the Q-sorting table (Figure 1) in advance and then asks the respondents to classify them [24].

The specific process is as follows. From 10 March 2021, to 25 April 2021, Q-sorting was conducted for 20 P-samples. Due to the COVID-19 pandemic, a non-face-to-face video conference (ZOOM) system was used with each P-sample. We explained the research background, purpose, and classification method to them in advance. In addition, after reading all Q statements, the participants were asked to place the most agreed with (+4) question and the most disagreed with (−4) question on both extremes. After that, positive, negative, and neutral questions were divided into three groups and placed in the positive (+3) to the negative (−3) questions.

Factor analysis was conducted with a Q package, that is, PQ Method (version 2.35) [28]. Afterwards, the centroid method (i.e., using the orthogonal varimax procedure) was applied [29]. For factor analysis (a technique for reducing a large number of variables into a smaller number of categories or factors), we attempted to detect the most appropriate number of factors by entering the number of factors from 7 to 2; then we extracted the factors with eigenvalues (EVs) of 1.00 or higher [25].

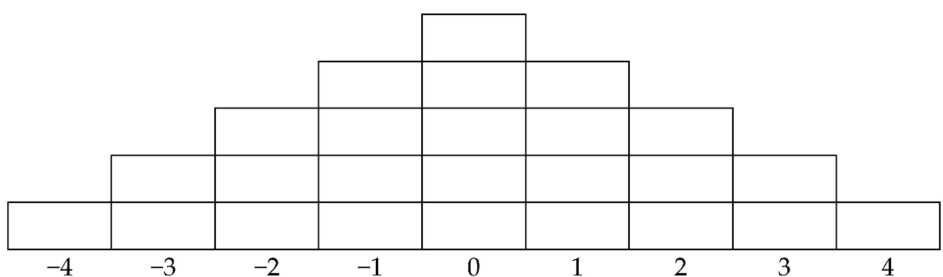

**Figure 1.** Q sorting table.

### 3. Results

### 3.1. Eigenvalues (EVs), Variance, and Correlations between Types

As a result of factor analysis, a total of five types of subjectivity were extracted. The EVs for each type were 3.89, 4.48, 3.96, 5.16, and 2.58, respectively, and the variance ratios were presented as 0.10, 0.22, 0.13, 0.33, and 0.09, respectively. The total variance was 0.87,

resulting in an explanatory power of 87% for all types. Furthermore, all factors met the Kaiser–Guttman criterion [17]. Table 3 yields the results.

**Table 3.** Eigenvalues (EVs) and variance between types.

|  | Type 1 | Type 2 | Type 3 | Type 4 | Type 5 |
|---|---|---|---|---|---|
| Eigenvalues (EVs) | 3.89 | 4.48 | 3.96 | 5.16 | 2.58 |
| % of explanatory variance | 0.10 | 0.22 | 0.13 | 0.33 | 0.09 |
| Total variance | 0.10 | 0.32 | 0.45 | 0.78 | 0.87 |

The correlation between the four types is shown in Table 4 below. Looking more closely, the correlation results were found to be 0.26 for Types 1 and 2, 0.46 for Types 1 and 3, 0.32 for Types 1 and 4, and 0.25 for Types 1 and 5, respectively. Moreover, the correlation was presented as 0.29 for Types 2 and 3, 0.30 for Types 2 and 4, 0.40 for Types 2 and 5, 0.52 for Types 3 and 4, 0.23 for Types 3 and 5, and 0.35 for Types 4 and 5. As can be seen from the results, the correlation between Types 3 and 4 was the highest, while Types 3 and 5 had the lowest.

**Table 4.** Correlations between types.

|  | Type 1 | Type 2 | Type 3 | Type 4 | Type 5 |
|---|---|---|---|---|---|
| Type 1 | 1 |  |  |  |  |
| Type 2 | 0.26 | 1 |  |  |  |
| Type 3 | 0.46 | 0.29 | 1 |  |  |
| Type 4 | 0.32 | 0.30 | 0.52 | 1 |  |
| Type 5 | 0.25 | 0.40 | 0.23 | 0.35 | 1 |

*3.2. Type 1: A Potent Means of Enhancing Lesson Concentration and Academic Performance*

Table 5 shows the Q-statement number and Z-scores of the results that belong to Type 1 that were positively or negatively recognized by the P-samples. In this type, the statements that participants positively agreed with were Q19, Q16, Q20, and Q17, with Z-values of 1.92, 1.81, 1.60, and 1.59, respectively. The respondents also had the most positive opinions for Q19 (Z-score = 1.92) and most negative perspectives for Q1 (Z-score = −2.00).

A total of four participants belonged to Type 1. Respondents of this type were positively aware of academic-related factors. In other words, they recognized that morning exercise had a positive effect on their learning. The P-sample number and the factor weight were P13 (0.90), P14 (0.88), P6 (0.68), and P7 (0.59).

**Table 5.** Statements with Z-scores of ±1.00 (or higher) from Type 1 to Type 5.

|  | Q-Statement Number | | Z-Score |
|---|---|---|---|
| Type 1 | Positive | 19 | 1.92 |
|  |  | 16 | 1.81 |
|  |  | 20 | 1.60 |
|  |  | 17 | 1.59 |
|  | Negative | 5 | −1.94 |
|  |  | 7 | −2.00 |
| Type 2 | Positive | 13 | 1.90 |
|  |  | 10 | 1.70 |
|  |  | 12 | 1.45 |
|  | Negative | 16 | −1.55 |
|  |  | 17 | −1.62 |
|  |  | 20 | −1.89 |

**Table 5.** *Cont.*

| | Q-Statement Number | | Z-Score |
|---|---|---|---|
| Type 3 | Positive | 6 | 1.95 |
| | | 8 | 1.65 |
| | | 7 | 1.31 |
| | Negative | 12 | −1.33 |
| | | 13 | −1.64 |
| | | 10 | −1.92 |
| Type 4 | Positive | 22 | 2.02 |
| | | 24 | 1.68 |
| | | 1 | 1.47 |
| | | 4 | 1.32 |
| | Negative | 7 | −1.42 |
| | | 8 | −1.89 |
| | | 5 | −1.96 |
| Type 5 | Positive | 25 | 1.98 |
| | | 14 | 1.80 |
| | | 15 | 1.50 |
| | | 13 | 1.20 |
| | Negative | 16 | −1.02 |
| | | 20 | −1.52 |
| | | 8 | −1.89 |

### 3.3. Type 2: Efficient Activities to Improve Physical Ability and a Healthy Body Image in Adolescence

For Type 2, the most positive Q statements were Q13 (Z-score = 1.90), followed by Q10 (Z-score = 1.70) and Q12 (Z-score = 1.45). Moreover, the most negative statement was Q20 (Z-score = −1.89). Table 5 displays the results and the Q statements numbers and Z-scores belonging to Type 2.

There were 5 participations in Type 2. These participants recognized that morning exercise not only improves physical aspects but also is efficient for acquiring a healthy body image. Participants P1, P8, P15, and P19 were included in Type 2, with factor weights of 0.75, 0.80, 0.84, and 0.68, respectively. Remarkably, the results indicate that participant P15 exhibited the highest factor weight, which represents this perspective well.

### 3.4. Type 3: The Driving Force behind Stable School Life and Sociability Development

Table 5 indicates that the most positive statement for participants in Type 3 was Q6 (Z-score = 1.95), followed by Q8 (Z-score = 1.65) and Q7 (Z-score = 1.31). In contrast, the most negative statement was Q10 (Z-score = −1.92), followed by Q13 (Z-score = −1.64), and Q12 (Z-score = −1.33).

A total of four P-samples belonged to Type 3, which was the same number of respondents with Types 1 and 2. Respondents P2, P9, P16, and P20 belonged to Type 3, with factor weights of 0.90, 0.92, 0.80, and 0.71, respectively. They were looking for the effects of morning exercise on social factors. This is to say, respondents were positively aware of factors such as close relationships with their friends, a sense of belonging to their school, and interest in school life.

### 3.5. Type 4: Building an Upright Character and Successful Changes in Daily Life

Table 5 illustrates the result for each Z-score and Type 4's statement numbers. The most positively viewed statement was Q22 (Z-score = 2.02), whereas the most negatively viewed statement was Q5 (Z-score = −1.96). Additionally, the other positive statements of respondents were Q24 (Z-score = 1.68), followed by Q1 (Z-score = 1.47) and Q4 (Z-score = 1.32). Conversely, the other negative statements were Q5 (Z-score = −1.96), followed by Q8 (Z-score = −1.89) and Q7 (Z-score = −1.42).

Five participants were identified as Type 4. As we have seen from Table 5, the respondents in this type were P3, P4, P10, P17, and P18, with a factor weight of 0.60, 0.59, 0.54, 0.64, and 0.89, respectively. Type 4 showed the largest number of respondents. The students included in this type were remarkably aware of factors such as patience, confidence, a challenging spirit, and bright personality. At the same time, these students acquired regular lifestyle habits through morning exercise.

### 3.6. Type 5: Raising Awareness of the Values of Participating in Sports and the Importance of Physical Activity

Table 5 presents the results and Z-scores of the statements belonging to Type 5. As clearly seen from the figures in this type, the statements that these P-samples positively agreed with are Q25, Q14, Q15, and Q13, with Z-scores of 1.98, 1.80, 1.50, and 1.20, respectively. In addition, the statements that participants disagreed with are Q16, Q20, and Q8, with Z-scores of −1.02, −1.52, and −1.89, respectively.

The smallest number of respondents were distributed in this type. All of Type 5's respondents were female, with a total of three people. Respondents of this type were discovering the value implied in sports and physical activities through morning exercise. The P-sample number and the factor weight were P5 (0.87), P11 (0.59), and P12 (0.79). The factor weight of participant P5 was the highest, which is a representation of this type of perspective.

### 3.7. Consensus Statements

Table 6 shows that the commonly agreed upon Q statements for P-samples of all types were Q24 (Z-score = 0.72) and Q25 (Z-score = 0.65). Proceeding from this result, two critical meanings can be presented. First, the majority of respondents experienced positive changes in their eating habits through the "0th period P.E. class". Second, it was a significant opportunity for accumulating knowledge for actively participating in physical activities, not only in adolescence but also in adulthood.

**Table 6.** Consensus statements.

| Q-Statement Number | Z-Score |
| --- | --- |
| 24 | 0.72 |
| 25 | 0.65 |

## 4. Discussion

We have discussed the EVs, variance, and correlation between the five types of student groups. We have also checked the results of a total of five types and their consensus statements.

In terms of each type of correlation, Types 3 and 4 (0.52) were the highest, followed by Types 1 and 3 (0.46) and Types 2 and 5 (0.40). This indicates that the respondents from Types 3 and 4 were responding positively to similar Q statements. Type 3 participants judged that school-life-related factors were the positive results of early morning exercise, and Type 4 participants positively recognized daily-life-related factors. The analysis of these results indicated that students' perception of life both in and outside of school is intimately linked to the physical activity in the "0th period P.E. class".

By contrast, the lowest correlation was revealed between Types 3 and 5. More specifically, it can be clearly seen that there is nothing in common between the positive and negative statements of the participants of Types 3 and 5. Namely, Type 3 respondents presented a higher awareness of the social relationships in school life. In contrast, Type 5 illustrated a higher recognition of the expansion of information and knowledge related to the value of physical activity.

We have also checked the results of a total of five types and their consensus statements. Based on the results, we note the following.

First, four respondents belonged to Type 1: a powerful means of enhancing lesson concentration and academic performance. Cotman et al. [30] argued that exercise helps form neural networks that smoothly supply blood and nutrients to the brain and connect brain neurons. In particular, it can be explained that cognitive ability was improved by increasing the Brain Derived Neurological Factor (BDNF). In addition, regular mid-intensity or higher aerobic exercise performed through early morning exercise not only positively changes brain neurons, but also improves physical strength and improves memory [31]. This study suggests the possibility that the "0th period physical education class" can positively affect brain activity and help improve cognitive and learning ability [32]. Therefore, as can be seen from the perception of Type 1 respondents, it can be interpreted that the "0th period physical education class" plays an indispensable role in students' studies. This is because, as mentioned earlier, high school students in the United States regularly performed high-intensity exercises that reached the maximum heart rate immediately after school for 16 weeks, and as a result, their concentration and performance in class improved significantly [33].

Second, four respondents fell under Type 2: efficient activities to improve physical ability and a healthy body image in adolescence. Park [34] conducted a study on high school students who participated in continuous morning exercise before regular class and then presented the following results. First of all, the fitness abilities rose quite a lot after the morning exercise classes. Secondly, physical self-concepts improved significantly. These findings prove Type 2 respondents' perceptions. Eventually, the participants experienced improvement in muscle strength, flexibility, and power through continuous physical education activities in the 0th period. Furthermore, it has become an activity to control weight, and, it has a significant meaning of maintaining a healthy body. This is due to the face that it significantly impacted the importance of body care [35]. As has been noted by a variety of scholars, it has been proven that there is a special correlation between experience in physical activity in adolescence and successful life in adulthood. As such, the experience of early morning exercise not only supports healthy adolescence, but it can also be a crucial means of leading to the experience of physical activity in adulthood.

Third, four respondents belonged to Type 3: the driving force behind a stable school life and sociability development. For this type, there was an outstanding positive perception of the factors related to the participants' school life. Middle school students' participation in physical education classes in the 0th period can lead to social development such as students' sociability and relationship with teachers [36], and long-term participation has a positive effect on their daily lives [13]. It has been proven that if students can participate in early morning exercise in the long term, it can be a significant opportunity for satisfying their school lives [37]. This is the basis for the "0th period physical education class" to be adopted as one of the critical educational policies that can be implemented by local offices of education in Korea.

Fourth, the greatest number of respondents fell under Type 4: building an upright character and successful changes in daily life. Out of 20 responders, 5 were classified as Type 4. These participants recognized that the "0th period physical education class" is an important class that can develop positive factors such as patience and confidence in daily life outside of school life. Moreover, respondents agreed that it is not limited to the physical aspect, but that there are many changes in the psychological aspect. In fact, the participation of middle school students in physical education activities in the 0th period is an essential variable influencing stress management from parents, studies, and friends [5]. Kim et al. [38] placed a heavy emphasis on the direct relationship between early morning physical activity and mental health. Thus, from the above perspective, it can be seen that the results of this study showed that early morning exercise for middle school students plays an indispensable role in their psychological stability [39].

Fifth, the lowest number of participants belonged to Type 5: raising awareness of the values of participating in sports and the importance of physical activity. Out of 20 P-samples, 3 were classified under Type 5. These respondents experienced new values with

regard to sports participation through early morning exercises. This finding needs to be noted by all teachers and leaders, including physical education teachers. The "0th period physical education class" should not be planned simply as a class, but needs to be approached in terms of essential physical activity in which all students participate [40]. This is because long-term participation in early morning exercise induced high school students to change their perception regarding physical activity positively [34].

Lastly, all respondents positively agreed with statements Q24 and Q25. Jeon et al. [23] suggested that constant Taekwondo training in adolescence is a useful means of acquiring healthy eating habits. For this reason, all respondents show a positive perception of statement Q24. Hence, continuous early morning exercise could also help improve students' eating habits. In addition, the "0th period physical education class" is an effective turning point for positively recognizing the value of physical activity [23]. In this respect, it is interpreted that all participants positively agreed to statement Q25.

## 5. Conclusions

This study has attempted to establish the subjective perception types and characteristics of Korean middle school students regarding participation in the "0th period physical education class". To achieve this purpose, the Q methodology was applied. The final 25 Q samples were selected through the composition of the Q-population; 20 Korean middle school students were selected as the P-sample on which Q-sorting was performed. For data analysis, centroid factor analysis and varimax rotation were performed using the PQ method program version 2.35. A total of 5 types were derived, and the total explanatory variance of all types was 87%. Type 1 (N = 4) was named "a powerful means of enhancing lesson concentration and academic performance". Type 2 (N = 4) has the theme of "efficient activities to improve physical ability and a healthy body image". Type 3 (N = 4) was named "the driving force behind stable school life and sociability development". Type 4 (N = 5) was described as "building an upright character and successful changes in daily life". And lastly, Type 5 (N = 3) was named "raising awareness of the values of participating in sports and the importance of physical activity". Furthermore, the consensus statements between each type were investigated as Q24 and Q25.

Lim et al. [5] suggested that elementary school students' participating in physical activity in the 0th period should pay attention to the educational meaning in that it created a new culture at the school. As such, there was a perception among the study participants that physical education classes in the 0th period create a different school sports culture from the existing physical education classes and activities. Therefore, according to the findings, it is believed that a serious judgment on the Korean physical education system is necessary. As we mentioned above, the "0th period physical education class" is not currently planned as a regular curriculum in the Korean education system. Therefore, the results of this study can be used as basic data for the implementation of a successful policy regarding the "breakthrough physical activity class". To discuss this in more detail, there should be a movement of policy changes in Korean physical education, centering on the Ministry of Education or the Regional Office of Education. For example, support for schools that conduct sports activities in the 0th period should be expanded. In addition, incentive policies for physical education teachers who run early morning exercise activities need to be enforced. This small move could lead to the implementation of a system that can establish itself as a formal class within all middle school curriculums in Korea.

The limitations of this study are as follows. First, in this study, since the type of perception for students was derived, it is necessary to study teachers' perceptions later. In fact, this is because the teacher is the one who runs the early morning exercise class. Second, in this study, students' perceptions were investigated through the Q methodology. In future research, it will be necessary to conduct research on students' perception through various qualitative or quantitative research methods.

**Author Contributions:** Conceptualization, W.J. and G.K.; methodology, W.J. and K.J.; investigation, W.J. and G.K.; data analysis, W.J.; writing—original draft preparation, W.J. and K.J.; writing—review

and editing, W.J., G.K. and K.J.; supervision, W.J., G.K. and K.J. All authors have read and agreed to the published version of the manuscript.

**Funding:** This research received no external funding.

**Institutional Review Board Statement:** The study was conducted according to the guidelines of the Declaration of Helsinki. Ethical review and approval were waived because necessary permissions were obtained from the schools (to which this study participants belonged) affiliated with this study.

**Informed Consent Statement:** Informed consent was obtained from all subjects involved in the study.

**Data Availability Statement:** The data presented in this study are available upon request from the corresponding authors.

**Conflicts of Interest:** The authors declare no conflict of interest.

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
