# Peer review of "Subjective Perceptions and Their Characteristics of Middle School Students Regarding the Effectiveness of the “0th Period Physical Education Class” in South Korea: The Q Methodology Application"

_sustainability, doi:10.3390/su132112081_

Round 1

Reviewer 1 Report

Title

Overall, the title of the paper is presented in detail. However, I am afraid that there are too many specific terms for the reader. For example, the reader might be not familiar what is the 0th period and what is the Q methodology. And because of several specific terms, the title is currently too long. Please consider rewording the title of the paper. Also, I see that in the title is only one time “, but I cannot find there the “ ends.

Abstract

I am still wondering what the meaning of “0th period physical education class” is.

I believe the version of program is not relevant information for the abstract.

I am wondering why a specific part of the text is in the italics.

Keywords

I am not sure that these are the best keywords. Also, there should not be overlap words used in the title and in the keywords. If there is overlap, then they keywords won’t additionally help users to find your research paper.

Introduction

Overall, the introduction reads well, and background information is presented in well detail.

The introduction starts with “more than 20 years ago”, and then Authors add “in particular, on December 20, 2011…”. This is about 10 years ago. So, what are the statistics for the statement “more than 20 years ago”?

The Authors shorten the term physical education as P.E, but on the first page line 37 is the only place where Authors use this shortening. Because of that, I cannot find this shortening any useful.

Please provide more specific description what is the "0th period physical education class" all about.

I would recommend Authors to state specific hypotheses of the study and also write paragraph entitled “the present study”.

Methods

Overall, the methods are clear and well presented to the reader.

Results

The results are also very detailed, I congratulate Authors for that.

Discussion

Overall, the discussion is well written, but it still needs some work. Please consider removing short paragraphs, especially the “one sentence paragraphs”.

Also, please add the paragraph strengths and limitations of the current study. Please also provide a specific paragraph entitled practical implications of the current study.

Conclusions

Overall, the conclusions are well written, but also the conclusions are too specific. Please try to avoid specific values and numbers in the conclusions section, because this should be only in the methods and results section.

Author Response

Dear reviewer:

First of all, I am deeply grateful for your thoughtful comments.

Abstract

I am still wondering what the meaning of “0th period physical education class” is.

I believe the version of program is not relevant information for the abstract.

I am wondering why a specific part of the text is in the italics.

Keywords

I am not sure that these are the best keywords. Also, there should not be overlap words used in the title and in the keywords. If there is overlap, then they keywords won’t additionally help users to find your research paper.

I really appreciate regarding reviewers' detailed comments. This part has been revised as your suggestion.

Introduction

Overall, the introduction reads well, and background information is presented in well detail.

The introduction starts with “more than 20 years ago”, and then Authors add “in particular, on December 20, 2011…”. This is about 10 years ago. So, what are the statistics for the statement “more than 20 years ago”?

The Authors shorten the term physical education as P.E, but on the first page line 37 is the only place where Authors use this shortening. Because of that, I cannot find this shortening any useful.

Please provide more specific description what is the "0th period physical education class" all about.

I would recommend Authors to state specific hypotheses of the study and also write paragraph entitled “the present study”.

I really appreciate regarding reviewers' detailed comments, too. This part has been revised as your opinion.

Discussion

Overall, the discussion is well written, but it still needs some work. Please consider removing short paragraphs, especially the “one sentence paragraphs”.

Also, please add the paragraph strengths and limitations of the current study. Please also provide a specific paragraph entitled practical implications of the current study.

Conclusions

Overall, the conclusions are well written, but also the conclusions are too specific. Please try to avoid specific values and numbers in the conclusions section, because this should be only in the methods and results section.

I really appreciate regarding reviewers' detailed comments, too. This part has been revised as your opinion.

Finally, the English part has been revised several times. Thank you.

As mentioned above, we will do our best to revise the paper and thank you ver much regarding your comments again.

best Regards,

Reviewer 2 Report

Thank you very for inviting me to review the current paper entitled “Subjective Perceptions and its Characteristics of Middle School Students Regarding the Effectiveness of “The 0th Period Physical Education Class in South Korea: The Q Methodology Application”. Wherein the concept of a 0th period physical education class in South Korea was analyzed using the Q-sort method.

Some recommendations are as follows:

  1. The paper needs to be language check – too many grammar mistake.
  2. Abstract should include some description of the Q methodology, since this is the focus of the paper besides the 0th Period Physical Education Class.
  3. Should relate the paper to sustainability – which is the main focus of the journal, current presentation better fitting for education sciences journal. Unless, the author can try to incorporate perhaps ESD 2030 goal to the background of the study.
  4. Research objectives can be presented in bullet form – why would physical education course be related to sustainability
  5. Could incorporate into the Q methodology introduction – Q sort or in contrast to Delphi method or Concept Mapping, or any other somewhat similar methodology used in categorizing participants’ viewpoints. Should also provide some previous related studies using Q methodology to further justify it usage.
  6. Line 276 “We have discussed the EVs, variance, and correlation between types”, not quite clear what this mean
  7. Discussion – sub heading needed for better clarity.
  8. Conclusion – similar with previous comments, should somehow relate to sustainability

In general, the paper seems interesting, physical education is very important. However, the paper needs to be grammar / language check, some clarification and expansion are also needed.

Author Response

Dear reviewer:

First of all, I am deeply grateful for your thoughtful comments.

  1. The paper needs to be language check – too many grammar mistake.

I really appreciate regarding reviewers' detailed comments. the English part has been revised several times. Thank you.

  1. Abstract should include some description of the Q methodology, since this is the focus of the paper besides the 0th Period Physical Education Class.

This part has been revised as your suggestion.

  1. Should relate the paper to sustainability – which is the main focus of the journal, current presentation better fitting for education sciences journal. Unless, the author can try to incorporate perhaps ESD 2030 goal to the background of the study.

  1. Research objectives can be presented in bullet form – why would physical education course be related to sustainability

I fully understand what you are referring to. P.E educattion means a powerful means of motivation to involve students how to participate and enjoy physical activity and then they will be able to learn the significance regarding Continuous physical activities for the near future. In this respect, I am confident that morning exercise will play a very important role in a successful transition to adulthood and can be linked to sustainability.

  1. Could incorporate into the Q methodology introduction – Q sort or in contrast to Delphi method or Concept Mapping, or any other somewhat similar methodology used in categorizing participants’ viewpoints. Should also provide some previous related studies using Q methodology to further justify it usage.

I really appreciate regarding reviewers' detailed comments, too. This part has been revised as your opinion.

  1. Line 276 “We have discussed the EVs, variance, and correlation between types”, not quite clear what this mean

This part has been revised as your suggestion.

  1. Discussion – sub heading needed for better clarity.
  2. Conclusion – similar with previous comments, should somehow relate to sustainability

I really appreciate regarding reviewers' detailed comments, too. This part has been revised as your suggestion.

As mentioned above, we will do our best to revise the paper and thank you ver much regarding your comments again.

best Regards,

Author Response

Dear reviewer:

First of all, I am deeply grateful for your thoughtful comments.

I really appreciate regarding reviewers' detailed comments.

I fully understand what you are referring to. P.E educattion means a powerful means of motivation to involve students how to participate and enjoy physical activity and then they will be able to learn the significance regarding Continuous physical activities for the near future. In this respect, I am confident that morning exercise will play a very important role in a successful transition to adulthood and can be linked to sustainability.

In summary, we tried to supplement the all part additionally according to your valuable point.

Accordingly, we have done our best to revise our manuscript and thank you very much.

best Regards,

Round 2

Reviewer 1 Report

Authors have done a well job by solving the issues raised by the Reviewers.

Author Response

I really appreciate regarding reviewers' detailed comments. Also, I would like to express my sincere gratitude for your evaluation.

Reviewer 2 Report

Thank you very much for letting me review the revised version of the paper entitled "Subjective Perceptions and their Characteristics of Middle School Students Regarding the Effectiveness of the “0th Period Physical Education Class” in South Korea: The Q Methodology Application. The author/s already made substantial improvement with regards to the methodology (which is much better now) and together with several improvements and clarification within the many sections of the paper. Just some minor comments: 1. limitations of the study - please add 2. any recommendations for policy improvements with regards to PE education 3. Discussion section - line 283 - the sentence "We have discussed the EVs, variance, and correlation between the 5 types of student groups", maybe use subsection instead eg. 4.1 EVs, variance, and correlation between the 5 types of student groups same with line 298 "We have also checked the results of a total of five types and their consensus statements", maybe 4.2 ..... 4. The author could also add in the discussions (or in the conclusion section) so what now? if these are the findings, what are the implications? hence, suggestions #2 In general, the author/s already done the needed revisions.

Author Response

Dear reviewer:

I am deeply grateful for your thoughtful review again.

  1. limitations of the study - please add

I really appreciate regarding reviewers' detailed comments. I added the content about limitations of my study. Thank you.

  1. any recommendations for policy improvements with regards to PE education

This part has been revised as your suggestion.

  1. Discussion section - line 283 - the sentence "We have discussed the EVs, variance, and correlation between the 5 types of student groups", maybe use subsection instead eg. 4.1 EVs, variance, and correlation between the 5 types of student groups same with line 298 We have also checked the results of a total of five types and their consensus statements", maybe 4.2 .....

I really appreciate regarding reviewers' detailed comments, too. This part has been revised as your opinion.

best Regards,

This manuscript is a resubmission of an earlier submission. The following is a list of the peer review reports and author responses from that submission.